# BPA Exposure Affects Mouse Gastruloids Axial Elongation by Perturbing the Wnt/β-Catenin Pathway

**DOI:** 10.3390/ijms25147924

**Published:** 2024-07-19

**Authors:** Paola Rebuzzini, Serena Rustichelli, Lorenzo Fassina, Ilaria Canobbio, Maurizio Zuccotti, Silvia Garagna

**Affiliations:** 1Laboratory of Biology and Biotechnology of Reproduction, Department of Biology and Biotechnology “Lazzaro Spallanzani”, University of Pavia, Via Ferrata 9, 27100 Pavia, Italy; maurizio.zuccotti@unipv.it (M.Z.); silvia.garagna@unipv.it (S.G.); 2Laboratory of Biochemistry, Department of Biology and Biotechnology “Lazzaro Spallanzani”, University of Pavia, Via Bassi 21, 27100 Pavia, Italy; serena.rustichelli@iusspavia.it (S.R.); ilaria.canobbio@unipv.it (I.C.); 3University School for Advanced Studies Pavia (IUSS), 27100 Pavia, Italy; 4Department of Electrical, Computer and Biomedical Engineering (DIII), University of Pavia, Via Ferrata 5, 27100 Pavia, Italy; lorenzo.fassina@unipv.it; 5Centre for Health Technologies (CHT), University of Pavia, Via Ferrata 5, 27100 Pavia, Italy

**Keywords:** Bisphenol A, gastruloids, axial elongation, Wnt/β-catenin, cadherins, pluripotency

## Abstract

Mammalian embryos are very vulnerable to environmental toxicants (ETs) exposure. Bisphenol A (BPA), one of the most diffused ETs, exerts endocrine-disrupting effects through estro-gen-mimicking and hormone-like properties, with detrimental health effects, including on reproduction. However, its impact during the peri-implantation stages is still unclear. This study, using gastruloids as a 3D stem cell-based in vitro model of embryonic development, showed that BPA exposure arrests their axial elongation when present during the Wnt/β-catenin pathway activation period by β-catenin protein reduction. Gastruloid reshaping might have been impeded by the downregulation of Snail, Slug and Twist, known to suppress E-cadherin expression and to activate the N-cadherin gene, and by the low expression of the N-cadherin protein. Also, the lack of gastruloids elongation might be related to altered exit of BPA-exposed cells from the pluripotency condition and their following differentiation. In conclusion, here we show that the inhibition of gastruloids’ axial elongation by BPA might be the result of the concomitant Wnt/β-catenin perturbation, reduced N-cadherin expression and Oct4, T/Bra and Cdx2 altered patter expression, which all together concur in the impaired development of mouse gastruloids.

## 1. Introduction

Bisphenol A (BPA), widely used in the production of polycarbonate plastics, epoxy resins and other polymeric materials [1,2], is one of the most diffused environmental toxicants (ETs) [3]. Because of its common presence in many items and cooking wares, the diet represents a primary source of contamination in humans [4,5,6,7]. BPA exerts endocrine-disrupting effects through estrogen-mimicking and hormone-like properties [8] with detrimental health impacts [7]. It increases the risk of some cancers [9] and impairs the cardiovascular system [10], neurogenesis [11] and gut microbiota [12,13]. As with many other ETs [14,15], BPA also negatively affects fertility and reproductive health [16,17,18]. Specifically, in females, it perturbs the hypothalamic–pituitary–gonadal axis, impairs oocyte maturation, reduces estradiol synthesis from the ovary and induces early pregnancy loss [19,20]; in male, it alters the genital apparatus and reduces sperm quality and production by altering hormone levels and/or miRNAs and proteins and sperm profiles [19,20,21].

BPA passes through the placenta, dramatically affecting embryonic development. In mice, treatment at environmentally relevant doses (1–100 μg/kg/day) causes delay in early embryo development [22] and increases apoptosis of blastocyst cells [22,23] and embryo implantation failure [24,25]. In vitro, during preimplantation development, a BPA dose of 5 or 25 µg/mL dysregulates the Hippo pathway and impedes cell division by disrupting the cytoskeletal organization during cell polarization and blastomeres’ compaction, all essential events for proper lineage segregation during segmentation [26].

BPA’s effects on early embryogenesis were also studied using mouse embryonic stem cells (mESCs) as an in vitro model of the blastocyst’s inner cell mass differentiation capacity. mESCs exposed to 0.04, 1.0 (the non-occupational and occupational exposure in humans, respectively), 25.0 or 100.0 µM BPA for 24 h showed mRNA and/or protein up-regulation of Oct-4, Sox2 and Nanog, together with a decrease of microRNA(miR)-134, an inhibitor of these three pluripotency markers [27]. Following differentiation into embryoid bodies, altered differentiation of the three germ layers was also observed [27]. Similarly, mESCs showed reduced cell viability and diminished neuronal (10.0 µM BPA) [28,29] or cardiac (10.0 to 263.2 μM BPA) [30,31] differentiation capacity.

Innovative stem-cell based models of embryo development [32,33] can profitably be used to address the lack of knowledge on the effects exerted by BPA during the peri-implantation stages, when the embryo is undergoing gastrulation, a process that entails an accurate and coordinated sequence of cell fate decisions and morphogenetic events.

Gastruloids, ESCs-derived 3 dimensional (3D) organoids that self-organize into free-floating globular aggregates [34,35,36,37], represent a powerful tool to study the early stages of mammalian post-implantation development in vitro [29,31,33]. They mimic, with a similar time course, well-organized gene expression domains delimiting the three body axes and axial elongation as it occurs in the mouse embryo at ~ embryonic day 6.0 (E) [34,35,37,38,39,40]. Specifically, in the mouse, the critical steps of gastrulation (E 6.25–9.5) are the anterior–posterior axis patterning and the specification of the germ layers. While the anterior epiblast will give rise to the ectoderm, mesoderm progenitors, expressing T/Brachyury (T/Bra), enter the primitive streak undergoing an epithelial-to-mesenchymal transition (EMT) required for their correct spatial relocation [41,42]. Endoderm progenitors move from the posterior epiblast epithelium while remodeling the basement membrane and downregulating apical–basal genes, and all of these features are reminiscent of a partial EMT program [41,43]. EMT is finely regulated by several signaling pathways, including Wnt/β-catenin, BMP and FGF [44,45,46,47,48,49], which coordinate the expression of specific genes that trigger the switch from E-cadherin to N-cadherin expression [50,51]. In gastruloids, as occurs in the embryo [34,52,53,54], the activation of the canonical Wnt/β-catenin signaling pathway through a Wnt agonist (CHIR 99021) prompts their elongation [34,37,38,39], determining their symmetry breaking and establishing antero–posterior polarity.

In this work, we took advantage of the 3D gastruloid cell culture platform to study the effects of BPA during gastruloid development, analyzing both the morphogenetic and molecular events involved in axial elongation. To this end, mouse gastruloids were exposed to seven increasing BPA concentrations (from 0.0001 to 50.0 µM) throughout the 120 h of differentiation or during the key events of (i) cell aggregation, (ii) a CHIR pulse of Wnt/β-catenin pathway activation, and (iii) axial elongation.

## 2. Results

### 2.1. BPA Interferes with Gastruloids’ Axial Elongation

In a first set of experiments, we confirmed that 0.01% DMSO, the vehicle used for BPA preparation, did not affect gastruloids axial elongation (*p* > 0.05), as previously shown [55]. Samples containing 0.01% DMSO were used and referred to as control (CTR) for all of the experiments described hereafter.

After aggregation (AA), at the end of the 120 h culture period, a major fraction (87%) of CTR gastruloids displayed an elongated shape (Figure 1A), whereas a minority (13%) remained as globular cell masses, as previously described with other mESC lines [38,39,56].

Then, the morphometric and morphological features of the shapes acquired by CTR gastruloids at the end of their differentiation were quantitatively evaluated using a specific algorithm (the gastruloid_morphology algorithm) developed by one of us (LF). Their elongated teardrop shape is quantitatively described by a median area of 1.9 × 10^5^ µm^2^ (morphometric feature) and median values of 1.9 for elongation and 0.73 for circularity (morphological features) (Figure 1C–E).

At the lowest BPA dose of 0.0001 µM, over 80% of the gastruloids showed median values of area (1.9 µm^2^), axial elongation (1.9) and circularity (0.69) that were not significantly different (*p* > 0.05) from the CTR group (Figure 1C–E). At BPA doses of 0.001 µM or higher, more than 95% of gastruloids displayed a globular and round shape (Figure 1B) with a decrease in elongation and an increase in circularity index median values to 1.3 and 0.8, respectively (Figure 1D,E).

Our results show that, when BPA was maintained throughout gastuloids’ culture, the 0.0001 µM dose, the lowest tested, did not induce any alteration in their morphological or morphometric parameters. The lowest effective dose that blocked elongation was determined at the concentration of 0.001 µM.

In the following experiments, BPA’s effects on gastruloids’ axial elongation were evaluated during the differentiation key events of (i) cell aggregation, (ii) a CHIR pulse of Wnt/β-catenin pathway activation, and (iii) axial elongation. To this end, samples were processed at different time points during culture where BPA was maintained for: (i) 48 h AA; (ii) 24 h (from 48 h to 72 h AA); (iii) 48 h (from 72 h to 120 h AA) (see Section 4).

### 2.2. BPA Interferes with CHIR Activation of the Wnt/β-Catenin Pathway

When exposed to BPA during the cell aggregation phase, 0–48 h AA, the gastruloids displayed axial elongation (Figure 2A,B) with a similar frequency (88%; *p* > 0.05) in the CTR and treated samples (ranging from 85 to 88%; *p* > 0.05). The gastruloids’ area (1.8–1.9 µm^2^), elongation (1.9–2.0) and circularity (0.85–0.9) (Figure 2C–E) values were comparable to those of the CTR (1.9 µm^2^ for area; 2.0 for elongation; 0.85 for circularity; *p* > 0.05) and within samples exposed to the different BPA doses (Figure 2). Similar results were obtained following BPA exposure during the axial elongation phase, from 72 h to 120 h AA. The majority of gastruloids were elongated (Figure 2K,L) and only a very low percentage (<3%; *p* > 0.05 compared to CTR) remained round in shape. The median area (1.9–2.1 µm^2^; Figure 2M), elongation and circularity (2.0–2.4 and 0.7–0.8, respectively; Figure 2N,O) values of the exposed gastruloids were not significantly different (*p* > 0.05), neither among the seven BPA doses nor compared to the CTR samples (1.9 µm^2^ for area; 1.9 for elongation; 0.8 for circularity; *p* > 0.05) (Figure 2).

When BPA was added during the CHIR treatment (Figure 2F,G), the 0.0001 µM dose did not affect gastruloids differentiation (Figure 2H–J). However, at higher doses, a significantly lower median area (1.1–1.2 × 10^5^ µm^2^; *p* < 0.05) and elongation (1.3–1.4; *p* < 0.05), along with higher circularity (0.9; *p* < 0.05) (Figure 2J), were observed compared to the CTR (Figure 2H–J). No significant differences (*p* > 0.05) were detected among samples exposed to 0.001–50 µM BPA.

These results confirmed that the differentiation of gastruloids is compromised at BPA concentrations of ≥0.001 µM, but only when exposure coincides with CHIR-induced activation of the Wnt/β-catenin pathway (0–120 h AA and 48–72 h AA).

For this reason, in subsequent experiments, cell cultures were exposed to BPA during the 48–72 h AA time frame only, and examined for:β-catenin protein expression, the key mediator of the Wnt signaling pathway [57]. Following translocation into the nucleus, β-catenin contributes to regulating the expression of Twist, Slug, and Snail genes [58,59,60], which are integral to the processes of EMT [61];the transcript levels of Twist, Slug, and Snail. Their transcription factors orchestrate the downregulation of E-cadherin and the upregulation of N-cadherin [62,63,64,65,66,67,68,69], two proteins that are involved in the adhesion and migration of embryonic cells [60];E-cadherin and N-cadherin expression. During development, reduced E-cadherin levels leads to decreased cell–cell adhesion and increased cellular motility. Increased N-cadherin levels are associated with enhanced cell migration and changes in cell adhesion properties, which are essential for EMT [60].

### 2.3. β-Catenin Is Lower in Gastruloids Exposed to BPA during CHIR Activation of the Wnt/β-Catenin Pathway

In CTR gastruloids, as expected, β-catenin protein quantity was higher following (72 h AA) than before CHIR activation (48 h AA) (Figure 3A,B). The concomitant exposure to CHIR and 0.0001 µM BPA did not alter protein expression (*p* > 0.05). However, its expression level was significantly (*p* < 0.05) lower at BPA concentrations from 0.001 to 50 µM, with fold changes ranging from 0.7 to 0.9 (Figure 3C,D). No significantly (*p* > 0.05) different protein quantity was detected among these latter BPA-exposed samples.

### 2.4. Snail, Slug and Twist Transcripts, as Well as N-Cadherin, but Not E-Cadherin, Protein Expression Are Lower in Gastruloids Exposed to BPA during CHIR Activation of the Wnt/β-Catenin Pathway

At 72 h AA, Snail, Slug and Twist transcripts were expressed at lower levels in BPA-exposed gastruloids compared to the CTR, with the exception of the 0.0001 µM dose (Figure 4). Specifically, the fold-change of Snail transcripts relative to the CTR was 0.8 at 0.001 µM, 0.5 at 0.01 and 0.1 µM, 0.3 at 1.0 and 10.0 µM BPA and 0.38 at 50.0 µM BPA. For Twist transcripts, the fold-change was 0.4 at 0.001, 0.01, 1.0 and 10.0 µM and 0.3 and 0.65 at 0.1 and 50 µM BPA, respectively. Slug displayed a markedly significant reduction, with a fold change of 0.5 at 0.001 µM, 0.1–0.18 from 0.01 to 10. 0 µM and 0.35 at 50 µM BPA. The expression of all three transcripts displayed a U-shaped trend, with a reduced number of transcripts at 0.01–10.0 µM BPA exposure, compared to the 0.001 or 50.0 µM BPA doses, which showed higher expression levels (Figure 4).

Both E-cadherin and N-cadherin are expressed in gastruloids at 72 h AA (Figure 5A), with the expression of N-cadherin recently reported by Suppinger and collaborators [70]. When compared to CTR samples, a slight, not significant (*p* > 0.05), reduction of E-cadherin quantity was evidenced following exposure to 0.0001–0.1 or 50 µM BPA doses, whereas at 1.0 or 10.0 µM doses its level displayed a mild, even not significant (*p* > 0.05), increase (Figure 5A,B). The E-cadherin protein level among all BPA exposed samples was not significantly different (*p* > 0.05).

At the lowest 0.0001 mM BPA exposure dose, N-cadherin expression was comparable to that of CTR samples (*p* > 0.05). Its expression was lower (*p* < 0.05; *p* < 0.001) when compared to all of the other BPA-exposed samples, with a fold change f 0.75, 0.6, 0.6 and 0.7 at 0.1, 1.0, 10.0 and 50.0 µM BPA, respectively (Figure 5A,B).

There was no difference (*p* > 0.05) in the N-cadherin protein level among all BPA dose samples, just mildly lower at the 1.0 and 10.0 µM doses.

The alteration of the parameters analyzed in the Wnt/β-catenin pathway and the maintenance of a rounded shape are suggestive of a lack of exit from pluripotency toward differentiation. This exit is tightly controlled [71,72,73] and the Wnt/β-catenin pathway contributes by inducing expression of the T/Brachyury [74] and Cdx2 [75] genes, whose combined expression is essential to establish the core signature of posterior axial progenitors, facilitating elongation and the definition of mesoderm and early endoderm territories, respectively, in embryos between E7.0 and E8.5 [76,77].

### 2.5. BPA Exposure during CHIR Activation of the Wnt/β-Catenin Pathway Inhibits Exit from Pluripotency and Interferes with Mesoderm and Endoderm Precursors Polarization

Oct4 immunofluorescence was bright and diffused in the round gastruloids of the CTR group at 72 h AA) (Figure 6B(B’)), while it was absent in the elongated gastruloids at 120 h AA (Figure 6F(F’)). Samples exposed to BPA during CHIR activation of the Wnt/β-catenin pathway exhibit a bright and diffused fluorescence at both 72 h AA (Figure 6D(D’)) and 120 h AA (Figure 6H(H’)).

The immunolocalization of T/Brachyury (T/Bra) and Cdx2, markers of mesoderm and endoderm progenitors, respectively, showed their polarized expression at the protruding tip of 120 AA CTR gastruloids (Figure 6J,L). Specifically, T/Bra positive cells are localized at the edge of the tip (Figure 6J(J’)), whereas Cdx2 positive cells extend from the tip toward the interior of the gastruloids (Figure 6L(L’)).

Round 120 h AA BPA-exposed gastruloids expressed both T/Bra and Cdx2 developmental markers, without any specific axial polarization (Figure 6N,P). T/Bra immunofluorescence extended toward the inner part of the cell mass (Figure 6N(N’)), whereas Cdx2 appeared more confined at the edge (Figure 6P(P’)).

These results indicate that the absence of axial elongation observed following 48–72 h AA BPA exposure parallels the maintenance of Oct4 expression and its immunolocalisation throughout the entire cell mass and lack of polarization of both T/Bra and Cdx2.

## 3. Discussion

In this study, we demonstrated that when gastruloids were exposed to BPA during the activation of the Wnt/β-catenin signaling pathway by CHIR, axial elongation was significantly inhibited. This elongation process is essential for symmetry breaking and the establishment of the antero–posterior polarity [34,53]. The underlying cause of this inhibition appears to be a disruption of the Wnt/β-catenin axis, specifically characterized by a reduction in the β-catenin protein levels. In the BPA-exposed gastruloids, the diminished quantity of β-catenin likely led to insufficient upregulation of critical genes such as Twist, Slug, and Snail. These transcription factors are crucial because they suppress E-cadherin and, in the case of Twist, activate N-cadherin genes [62,63,64,65,66,67,68,69].

Our analysis revealed that the levels of E-cadherin remained comparable between the control and BPA-exposed samples, while the upregulation of N-cadherin was notably low in the BPA-exposed gastruloids. This imbalance likely interferes with cell movements, thereby inhibiting axial elongation and disrupting the formation and relative localization of the germ layers. Specifically, this disruption is particularly evident in mesoderm precursor cells, where the upregulation of N-cadherin normally prevails over E-cadherin following the activation of the EMT program [41,61]. N-cadherin upregulation also occurs in mesendoderm and definitive endoderm cells, where its co-expression with E-cadherin regulates cell adhesion for proper cell segregation [41]. Previous studies [78,79] have shown that N-cadherin depletion results in defective gastrulation in the mouse. Moreover, E-cadherin expression promotes the translocation of β-catenin out of the nucleus, contributing to the suppression of the Wnt/β-catenin pathway [80].

This intricate interplay of cadherin expression and β-catenin localization underscores the delicate balance required for proper embryonic development and how BPA exposure disrupts this balance, leading to significant developmental defects.

The reduced quantity of β-catenin protein in BPA-exposed samples may also negatively impact the progressive exit of cells from the pluripotent state and their differentiation. These coordinated events are essential for the proper elongation and development of gastruloids [70]. In fact, the round-shaped BPA-exposed gastruloids at the end of the culture period still exhibited high expression of the Oct4 pluripotency marker in almost all aggregated cells, together with diffused expression of T/Bra and Cdx2 germ layers markers. During gastrulation, the dynamic changes of Oct4 expression levels play a pivotal role in controlling early cell fate decisions, connecting the transition from pluripotency to germ layers specification during mammalian development [81,82,83]. In early developmental stages, Oct4 exerts two counteracting actions: it suppresses the mesoderm-inducing Wnt/β-catenin signaling pathway and provides competence to the T/Bra gene to respond to Wnt/β-catenin signaling [84]. Additionally, in the embryo between E7.0 and E8.5, the activated Wnt/β-catenin pathway positively regulates the expression of both T/Bra [74] and Cdx2 [75] genes. Their combined expression is essential to establish the core signature of posterior axial progenitors, permitting embryo elongation and the definition of mesoderm and early endoderm territories, respectively [76,77].

Exposure to BPA between 48 h and 72 h AA of gastruloid differentiation, in concomitance with CHIR induction, might interfere with the transition of cells from pluripotency to differentiation. This interference disrupts the balance between Oct4 levels and the regulation of differentiation markers by Wnt/β-catenin activation. It was previously demonstrated that BPA and other ETs (e.g., xenoestrogens) can induce an up-regulation of pluripotency genes (Nanog and Oct4) at the mRNA and/or protein levels in mouse pluripotent cells and their derived embryoid bodies [27,85,86]. Since BPA, mimicking estrogen’s action, can bind to non-classical nuclear receptors [87,88], it is possible that the high Oct4 levels that we observed following BPA exposure are due to its estrogenic activation of the estrogen-related receptor beta (esrrb). The nuclear orphan receptor esrrb plays crucial roles in early mouse development, particularly in pluripotent cells where it induces the expression of genes involved in self-renewal and maintains the stability of the pluripotency network [89,90]. The molecular mechanisms of BPA’s influence during gastruloid formation and its potential interaction with esrrb require further study. Both traditional methods, like in vitro binding assays and modulation of esrrb expression, and computational approaches, such as molecular docking and structure-based virtual screening, can help predict and confirm the dynamic behavior and interactions of esrrb–BPA complexes.

Together, these approaches form a comprehensive strategy to understand the molecular mechanics at play. They will contribute to deepening the knowledge on how esrrb and BPA might interact, thereby shedding light on the potential impacts of environmental chemicals on embryonic development and the stability of crucial developmental pathways.

## 4. Materials and Methods

### 4.1. Cell Lines

HM1wt mouse embryonic stem cells (mESCs) and mouse STO-SNL2 cells (STO, American Type Culture Collection CRL-2225) were cultivated as previously described [91]. Briefly, HM1wt mESC were grown in Knockout DMEM supplemented with 15% qualified fetal bovine serum (FBS), 2 mM L-glutamine, 1× non-essential amino acids, 0.5% penicillin/streptomycin (all from Thermo Fisher Scientific, Waltham, MA, USA), 0.1 mM β-mercaptoethanol (Merck Millipore, Burlington, MA, USA) and 500 U/mL ESGRO-LIF (Merck) (LIF medium). STO cells were maintained in DMEM (Merck) supplemented with 10% FBS, 4 mM L-glutamine, 1× non-essential amino acids, 0.5% penicillin–streptomycin solution (all from Thermo Fisher Scientific), 0.1 mM β-mercaptoethanol and 0.2 mg/mL geneticin (Merck Millipore, Burlington, MA, USA). ESCs were routinely passaged enzymatically every 2–3 days with 0.05% trypsin/EDTA (Thermo Fisher Scientific, Waltham, MA, USA), alternating one passage on STO feeder cells with two passages on gelatin-coated p55 dishes (Corning, New York, NY, USA), and maintained in an incubator at 37 °C with 7.5% CO_2_ in air.

### 4.2. BPA Preparation

BPA powder (Merck) was dissolved in 100% DMSO (Merck Millipore, Burlington, MA, USA) to a stock concentration of 100 mM. The solution was maintained at 4 °C to ensure the stability of the molecule.

### 4.3. Gastruloid Formation and BPA Treatments

Gastruloids were obtained following the protocol published on Protocol Exchange (doi: https://doi.org/10.1038/protex.2018.094) and summarized in Figure 7A.

Briefly, HM1wt cells were grown in 2iLIF medium [LIF medium, 3 μM CHIR 99021, 1 μM PD0325901 (Tocris Biosciences, Bristol, UK)] for 24 h (Figure 1A). Then, they were gently rinsed with 1× PBS and enzymatically detached with 0.05% trypsin/EDTA, centrifuged and counted. Three hundred mESCs in 40 μL of N2B27 medium [Neurobasal (Thermo Fisher Scientific, Waltham, MA, USA) and DMEM-F12 (Merck) 1:1 *v*/*v*, 1× N2 (Thermo Fisher Scientific, Waltham, MA, USA), 1× B27 (Thermo Fisher Scientific, Waltham, MA, USA), 0.2 mM glutamine (Thermo Fisher Scientific, Waltham, MA, USA) and 0.1 mM β-mercaptoethanol] per well were seeded in U-bottomed 96-well plates (Thermo Fisher Scientifics, Waltham, MA, USA) up to 120 h. Between 48 h and 72 h, the cells were exposed to 3 µM CHIR 99021 in N2B27 (Figure 7A).

BPA was administered at the concentrations of 0.0001, 0.001, 0.01, 0.1, 1.0, 10.0 and 50.0 µM, according to previous published data [27,28,86], at different time points during the gastruloids’ culture and maintained for: (i) 120 h (from seeding up to the end of culture); (ii) 48 h after aggregation (AA); (iii) 24 h (from 48 h to 72 h AA); and (iv) 48 h (from 72 h to 120 h AA) (Figure 7B). BPA was added fresh at each medium change (Figure 7B, gray arrowheads).

In parallel, gastruloids were differentiated in the presence of the vehicle 0.01% DMSO (CTR).

At least three independent experiments were performed.

At the end of the culture period (120 h AA), a total of 300 CTR and 45 gastruloids for each BPA concentration were collected, observed with an Olympus BX60 microscope (10× objective) and images captured with a DP72 camera (Olympus, Shinjuku, Tokyo, Japan) using cellSens 1.4.1 software for the following morphology analyses.

### 4.4. Gastruloid Morphology

Starting from the SOTA algorithm [92], a “*gastruloid_morphology*” custom-made script, written in MATLAB^®^ Programming Language (Release R2022b, The MathWorks, Inc., Natick, MA, USA), was employed to quantify the area (morphometric feature), circularity and elongation (morphological features) of each single gastruloid.

#### 4.4.1. Morphometric Features

The boundary of each CTR or BPA-treated gastruloid was delimited to determine its area, along with the minor and major axes length.

#### 4.4.2. Morphological Features

The shape of each single gastruloid was defined by the following parameters:(1)Circularity, comprised between 0 and 1, where the value 0 represents an ellipse, whereas the value 1 represents a circumference;(2)Elongation is a form factor; an elongation equal to 1 represents a circumference, whereas an elongation >1 represents an ellipse.

### 4.5. Protein Extraction, Electrophoresis and Immunoblotting

Pools of 12 or 24 gastruloids for CTR or BPA-exposed samples, respectively, were pelleted at 1000× *g* for 5 min. Cells were lysed in 60 µL of RIPA buffer (2% Nonidet P-40, 1% sodium deoxycholate, 0.2% SDS, 4 mM PMSF, 100 µg/mL leupeptin, 100 µg/mL aprotinin, 4 mM sodium orthovanadate, 10 mM HEPES, 137 mM NaCl, 2.9 mM KCl, 12 mM NaHCO_3_, pH 7.4; all from Sigma-Aldrich, St. Louis, MO, USA) under gentle rotation for 30 min at 4 °C. Lysates were then centrifuged at 18,000× *g* for 1 min, the supernatants collected and maintained at −80 °C until used.

For immunoblotting analysis, 30 µL of SDS-sample buffer 3X (37.5 mM TRIS, 288 mM glycine, 6% SDS, 1.5% DTT, 30% glycerol, and 0.03% bromophenol blue and 3% 2-mercaptoethanol, pH 8.3) was added to the protein samples and boiled for 3 min at 96 °C.

Then, 40 µL of each sample was separated on 7.5% SDS-PAGE electrophoresis and transferred to a PVDF membrane. Membranes were blocked with 5% BSA (SERVA, Heidelberg, Germany) in TBS (20 mM Tris, 500 mM NaCl, pH 7.5) for 2 h at room temperature and incubated with primary antibodies (Table 1), diluted in TBS supplemented with 5% BSA, 0.1% Tween 20 and 0.02% NaN_3_ overnight at 4 °C under gentle agitation. Then, membranes were extensively washed with TBS containing 0.1% Tween 20 on agitation. Membranes were then incubated with the appropriate HRP-conjugated secondary antibodies (Table 1) for 1 h at room temperature. Repeated washings were performed with PBS. Proteins were visualized with Millipore Immobilon chemiluminescence reaction (Merck). Images of the staining were acquired with the ChemiDoc XRS system (Bio-Rad, Hercules, CA, USA). Densitometric intensities of the bands were determined by using Quantity One-4.6.8 software (Bio-Rad, Hercules, CA, USA).

At least four independent experiments were performed.

### 4.6. RNA Extraction, Reverse Transcription and Quantitative Real-Time PCR

TRIzol (Thermo Fisher Scientific, Waltham, MA, USA) was used to extract RNA from pools of 20 gastruloids according to the manufacturer’s instructions. Six independent experiments were performed and a total of 120 gastruloids each were analyzed for the CTR and each BPA-exposed condition.

Reverse transcription was performed in a final volume of 20 µL reaction mixture with 300 ng RNA, 1× PCR buffer, 5 mM MgCl_2_, 4 mM of each dNTP, 0.625 µM oligo d(T)16, 1.875 µM Random Hexamers, 20 U Rnase Inhibitor, 50 U MuLV reverse transcriptase (all from Thermo Fisher Scientific, Waltham, MA, USA) at 25 °C for 10 min, 42 °C for 15 min, 99 °C for 5 min. One twentieth of the resulting cDNA was amplified in duplicate by Real-Time PCR in 20 µL reaction mixture with 200 nM of each specific primer (designed using Primer 3 software; see Appendix A) and the MESA GREEN qPCR Master Mix Plus for SYBR assay no ROX sample (Eurogentec, Seraing, Belgium) at 1× final concentration. The amplification reaction was carried out in a Rotorgene 6000 (Qiagen, Hilden, Germany) at 95 °C for 5 min, followed by 40 cycles at 95 °C for 10 s, 60 °C for 15 s, 72 °C for 20 s. β-2-microglobulin gene expression was used for normalization [93]. The comparative concentration analysis was completed with Rotorgene 6000 Series Software 1.7.

### 4.7. Immunofluorescence

Three-dimensional (3D) immunofluorescence on gastruloids was performed according to a protocol previously described [38], with a few modifications. Briefly, a total of 20 gastruloids of CTR and 20 gastruloids exposed to 0.001 or 0.01 µM BPA were washed 3 times for 10 min at RT with PBS, then with PBS/10% FBS/0.5% Triton X-100 (PBSFT) 3 times for 10 min and finally with PBSFT for 1 h at 4 °C on a low-speed orbital rocker. Gastruloids were then incubated with the specific primary antibodies (Table 1) in PBSFT for 48 h at 4 °C on a low-speed orbital rocker. Following washes with PBSFT, the aggregates were incubated overnight with the secondary primary antibody (Table 1) in PBSFT at 4 °C on a low-speed orbital rocker. Nuclei were counterstaining for two hours with DAPI.

Images were obtained using a 20× objective of a confocal Leica SP8 light microscope equipped with a motorized XY scanning stage controlled by the LAS X Navigator stitching Lite version software (Leica, Wetzlar, Germany).

### 4.8. Statistics

Western blotting and qRT-PCR data are presented as mean ± standard deviation (SD), while the morphometric and morphological parameters are expressed as mean ± 95% confidence interval for the difference between means. Data were analyzed by one-way ANOVA and by the post hoc LSD test (selecting a significance level of 0.05) using SigmaStat v4.0 software.

## 5. Conclusions

In conclusion, we exploited gastruloids as a 3D stem cell-based in vitro model of early embryonic development to investigate BPA-induced perturbations during the peri-implantation period, within E5.5 to E8.5, when the definition of the three germ layers and the anteroposterior axis occurs. For the first time, we showed that BPA exposure prompts a block of gastruloids’ elongation, likely due to impaired activation of the Wnt/β-catenin pathway, which in turn impacts cell-to-cell junctions and the exit of cells from the pluripotency condition for proper differentiation. In addition, our results support previous observations that indicated that the 48–72 h AA period is the most crucial phase for both elongation and correct patterning of mouse gastruloids.

## Figures and Tables

**Figure 1 ijms-25-07924-f001:**
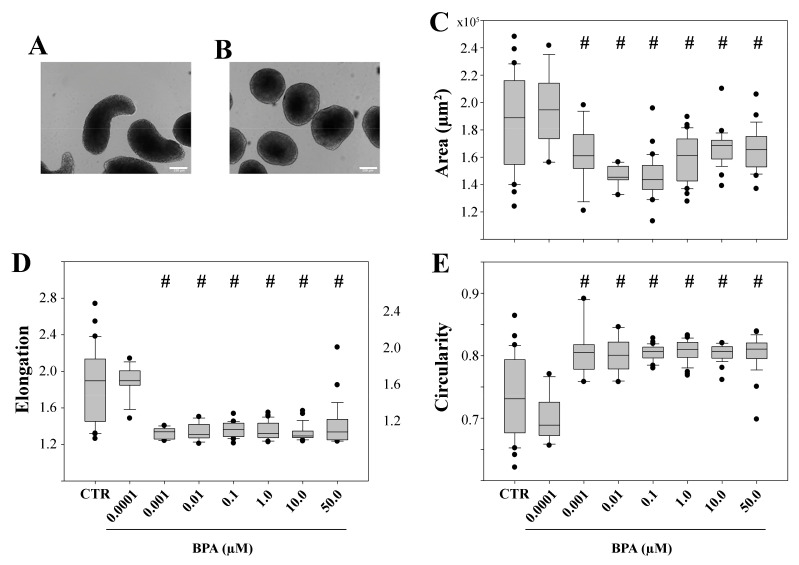
Representative bright-field images at 120 h after aggregation of CTR (**A**) and 0.01 µM BPA-exposed (**B**) gastruloids; magnification: 100×; bar: 200 µm. Quantification of area (**C**), elongation (**D**) and circularity (**E**) parameters. # *p* < 0.05 when compared to CTR. The horizontal line within the box is the median value; the upper and lower lines are the 75th percentile and 25th percentile, respectively. Black dots are outlier values.

**Figure 2 ijms-25-07924-f002:**
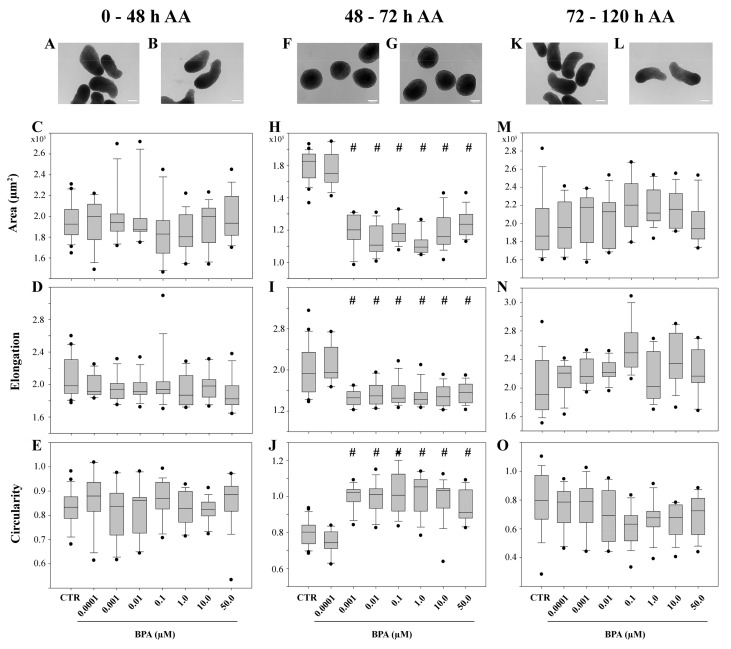
Representative bright-field images of BPA-exposed gastruloids at 0–48 h after aggregation [AA; (**A**,**B**)], 48–72 h AA (**F**,**G**) and 72–120 h AA (**K**,**L**); AA, after aggregation. Magnification: 100×; Bar: 200 µm. Quantification of area (**C**,**H**,**M**), following BPA exposure 0–48 h AA), elongation (**D**,**I**,**N**), following BPA exposure 48–72 h AA) and circularity (**E**,**J**,**O**), following BPA exposure 72–120 h AA). # *p* < 0.05 when compared to CTR. The horizontal line within the box is the median value, whereas the upper and lower lines are the 75th percentile and 25th percentile, respectively. Black dots are outlier values.

**Figure 3 ijms-25-07924-f003:**
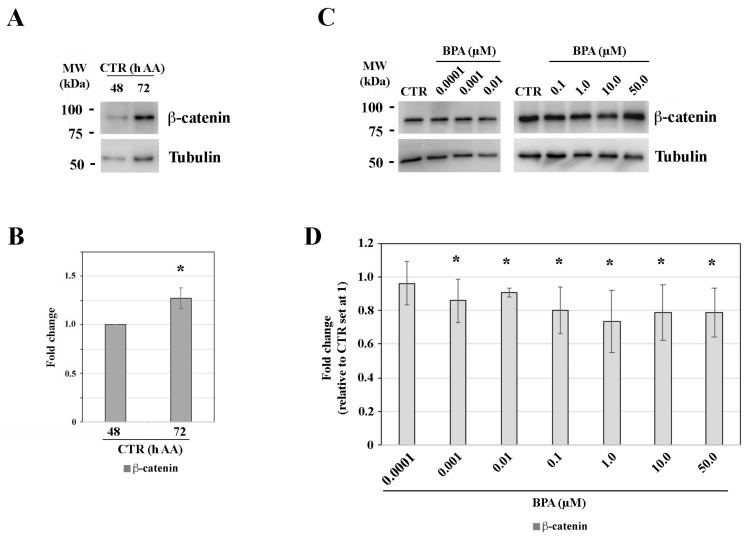
Representative western blotting of (**A**) β-catenin expression in CTR gastruloids collected at 48 h AA and 72 h AA and (**B**) their quantitation (CTR 48 h AA expression was set at 1). Representative western blotting of (**C**) β-catenin at 72 h AA and (**D**) its quantitation (CTR 72 h AA expression was set at 1). Tubulin protein expression was used as a loading control. * *p* < 0.05.

**Figure 4 ijms-25-07924-f004:**
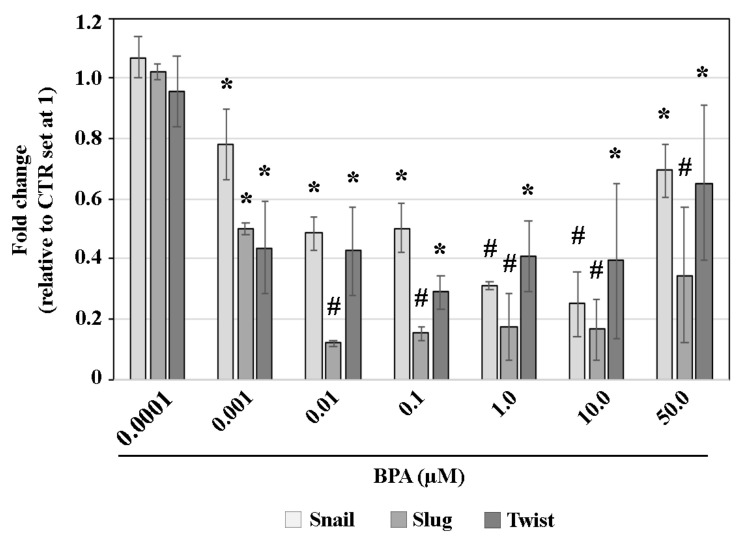
Fold change, relative to CTR, of the expression profile of Snail, Slug and Twist genes at 72 h AA, following 48–72 h AA BPA exposure. * *p* < 0.05; # *p* < 0.001.

**Figure 5 ijms-25-07924-f005:**
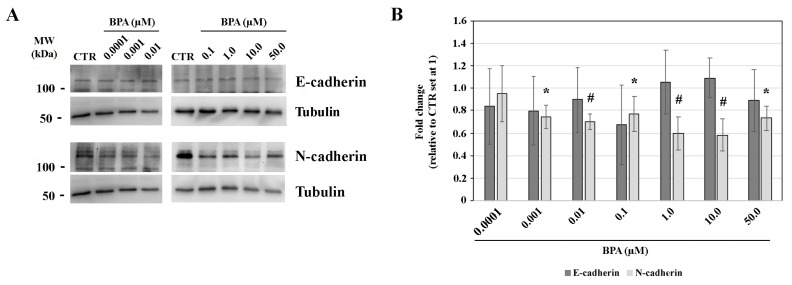
Representative western blotting of (**A**) E-cadherin and N-cadherin proteins at 72 h AA, following 48–72 h AA BPA exposure, and (**B**) the results of their quantitation. Tubulin protein expression is used as a loading control. * *p* < 0.05; # *p* < 0.001.

**Figure 6 ijms-25-07924-f006:**
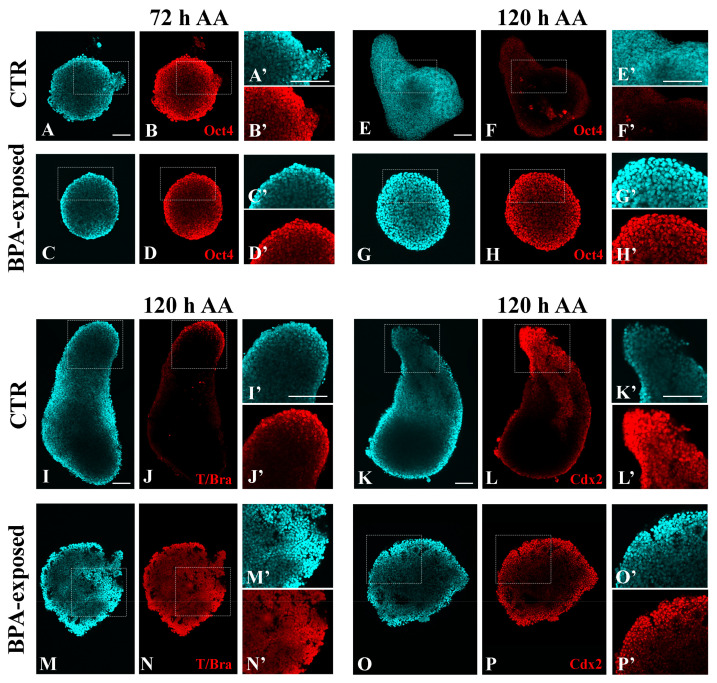
Representative confocal images of CTR and 48–72 h AA BPA-exposed (0.001 µM) gastruloids showing immunostaining for Oct4 (**B**,**D**,**F**,**H**), T/Bra (**J**,**N**) and Cdx2 (**L**,**P**) (Bar: 100 μm). Magnification of Oct4 (**B’**,**D’**,**F’**,**H’**), T/Bra (**J’**,**N’**) and Cdx2 (**L’**,**P’**) red signals in the insets (Bar: 200 μm). Nuclei were counterstained with DAPI (**A**,**C**,**E**,**G**,**I**,**K**,**M**,**O**; Magnification **A’**,**C’**,**E’**,**G’**,**I’**,**K’**,**M’**,**O’**).

**Figure 7 ijms-25-07924-f007:**
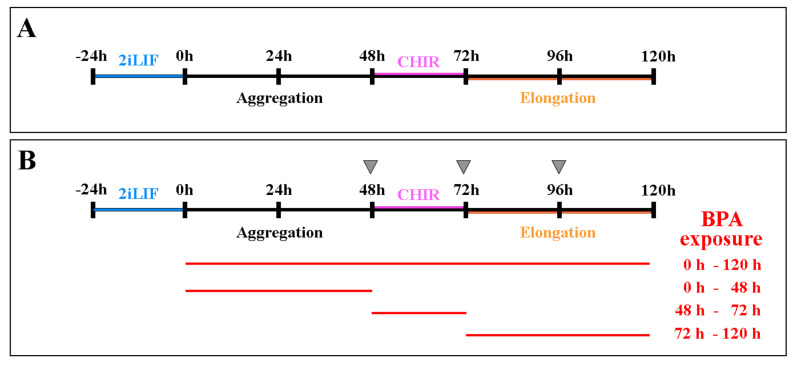
(**A**) Experimental design of mouse gastruloids’ formation; (**B**) BPA exposure timing during gastruloids’ formation. Gray arrowheads represent the timepoints of medium change. CHIR, CHIR 99021.

**Table 1 ijms-25-07924-t001:** Primary and secondary antibodies used, their source and dilution.

Antibody	Source	Catalogue Number	Dilution
Rabbit anti-human β-catenin	Abcam, Cambridge, UK	32572	1:1000
Rabbit anti-human phospho-β-catenin (Tyr654)	ECM Biosciences, Versailles, KY, USA	4021	1:1000
Rabbit anti-phospho-Gsk-3α/β (Ser21/9)	Cell Signaling Technology, Danvers, MA, USA	BK9331S	1:1000
Rabbit anti-phospho-Akt (Ser473)	Cell Signaling Technology, Danvers, MA, USA	9275	1:1000
Rat anti-human E-cadherin	Abcam, Cambridge, UK	ab11512	1:1000
Rabbit anti-human N-cadherin	Abcam, Cambridge, UK	12221	1:1000
Rabbit anti-brachyury	Cell Signaling Technology, Danvers, MA, USA	81694	1:500
Rabbit anti-cdx2	Cell Signaling Technology, Danvers, MA, USA	3977	1:100
Rabbit anti-oct4	Cell Signaling Technology, Danvers, MA, USA	2840	1:300
Mouse α-tubulin	Santa Cruz, Santa Cruz, CA, USA	sc-32293	1:1000
Anti-mouse HRP	Sigma-Aldrich, St. Louis, MO, USA	SAB5300168	1:2000
Anti-rat HRP	Sigma-Aldrich, St. Louis, MO, USA	A9037	1:5000
Anti-rabbit HRP	Sigma-Aldrich, St. Louis, MO, USA	A6154	1:5000
Anti-rabbit Alexa 555	Thermo Fisher Scientific, Waltham, MA, USA	A-21428	1:500

## Data Availability

Data are contained within the article and Appendix A.

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
