# Peer review of "BPA Exposure Affects Mouse Gastruloids Axial Elongation by Perturbing the Wnt/β-Catenin Pathway"

_ijms, 2024, doi:10.3390/ijms25147924_

Round 1

Reviewer 1 Report

Comments and Suggestions for Authors

The authors evaluate the effect of BPA exposure on mouse gastruloid elongation, as a model for potential effect of BPA exposures on peri-gastrulation stage embryos. They employ 7 different concentrations, and expose the gastruloids either throughout the 120 hrs of their development, between 0-48 hrs, 48-72 hrs (during the CHIR pulse) or 72-120 hrs. 

The manuscript is well written, providing systematic evidence for the effects of BPA on early mouse gastruloid development.

The introduction is very well written, providing motivation and sufficient background, including relevant previous results.

The authors find that even 0.001uM of BPA (well bellow toxicity levels) is sufficient to extinguish gastruloid elongation. When dissecting the time window at which this effect takes effect, it appears the CHIR pulse period (48-72 hrs) is the one susceptible to the BPA effect.

They then evaluate the effects of BPA on different Wnt activation indicators (bCat protein levels, twist/snail/slug mRNA levels, Ecad/Ncad expression. All these indicators suggest, again, that from 0.001uM of BPA, the activation of Wnt and its downstream effects are repressed.

Finally, the results are further corroborated with immunostainings for Oct4, Brachyury, Cdx2, showing BPA exposure leads to later maintenance of Oct4 expression, and diffuse expression of Brachyury at 120hrs. Immunostains for Ecad/Ncad would have been interesting (and complementary to their western blots), but this is not a must.

The discussion is somewhat weighed more towards the developmental interpretation of the results (Eca/Ncad expression during gastrulation), but only lightly touches the potential mechanism of action (through esrrb) – this could be expanded a bit, potentially suggesting what to study to corroborate this mechanism (e.g. manipulations of esrrb, etc.)

Minor comments:

Line 172: Instead -> However

Line 265: deplition -> depletion

274: of of

276: Cxd2 -> Cdx2

Comments on the Quality of English Language

The paper is very well written.

Author Response

Reviewers’ Comments

Q, your questions; A, our answers (our replies in italics).

Reviewer 1

The authors evaluate the effect of BPA exposure on mouse gastruloid elongation, as a model for potential effect of BPA exposures on peri-gastrulation stage embryos. They employ 7 different concentrations, and expose the gastruloids either throughout the 120 hrs of their development, between 0-48 hrs, 48-72 hrs (during the CHIR pulse) or 72-120 hrs. 

The manuscript is well written, providing systematic evidence for the effects of BPA on early mouse gastruloid development.

The introduction is very well written, providing motivation and sufficient background, including relevant previous results.

The authors find that even 0.001uM of BPA (well below toxicity levels) is sufficient to extinguish gastruloid elongation. When dissecting the time window at which this effect takes effect, it appears the CHIR pulse period (48-72 hrs) is the one susceptible to the BPA effect.

They then evaluate the effects of BPA on different Wnt activation indicators (bCat protein levels, twist/snail/slug mRNA levels, Ecad/Ncad expression. All these indicators suggest, again, that from 0.001uM of BPA, the activation of Wnt and its downstream effects are repressed.

We thank the reviewer for these words of appreciation

Q1: Finally, the results are further corroborated with immunostainings for Oct4, Brachyury, Cdx2, showing BPA exposure leads to later maintenance of Oct4 expression, and diffuse expression of Brachyury at 120hrs. Immunostains for Ecad/Ncad would have been interesting (and complementary to their western blots), but this is not a must.

A1: We thank the reviewer for this suggestion. We agree that “immunostains for Ecad/Ncad would have been interesting (and complementary to their western blots)”. However, as also stated by this reviewer, “this is not a must” as the results obtained through the Western blotting analyses for both E- and N-cadherin are very informative and do not require, in our opinion, other complementary experiments. 

Q2: The discussion is somewhat weighed more towards the developmental interpretation of the results (Eca/Ncad expression during gastrulation), but only lightly touches the potential mechanism of action (through esrrb) – this could be expanded a bit, potentially suggesting what to study to corroborate this mechanism (e.g. manipulations of esrrb, etc.)

A2: We thank the reviewer for this suggestion. The final part of the Discussion is now expanded, introducing the following paragraph:

“The nuclear orphan receptor esrrb plays crucial roles in early mouse development, particularly in pluripotent cells where it induces the expression of genes involved in self-renewal and maintains the stability of the pluripotency network [90,91]. The molecular mechanisms of BPA's influence during gastruloid formation and its potential interaction with esrrb require further study. Both traditional methods, like in vitro binding assays and modulation of esrrb expression, and computational approaches, such as molecular docking and structure-based virtual screening, can help predict and confirm the dynamic behavior and interactions of esrrb-BPA complexes.

Together, these approaches form a comprehensive strategy to understand the molecular mechanics at play. They will contribute to deep the knowledge on how esrrb and BPA might interact, thereby shedding light on the potential impacts of environmental chemicals on embryonic development and the stability of crucial developmental pathways.” (lines 349-361, pages10-11). 

Q3: Minor comments

Line 172: Instead -> However

Line 265: deplition -> depletion

274: of of

276: Cxd2 -> Cdx2

A3: Done.

Comments on the Quality of English Language

The paper is very well written.

We thank the reviewer for these words of appreciation.

Reviewer 2 Report

Comments and Suggestions for Authors

The main aim and question of this manuscript is to use the 3D gastruloids cell culture platform to study the effects of BPA during gastruloids development, analyzing both the morphogenetic and molecular events involved in their axial elongation.

This manuscript shows rich content, providing a deep insight for some works: the study is within the journal’s scope, and I found it to be well-written, providing sufficient information. Even if the manuscript provides an organic overview, with a densely organized structure and based on well-synthetized evidence, there are some suggestions necessary to make the article complete and fully readable. For these reasons, the manuscript requires major changes.

Please find below an enumerated list of comments on my review of the manuscript:

MINOR POINTS:

The authors should provide a list of the abbreviations, mentioned in this manuscript.

MAJOR POINTS:

INTRODUCTION:

LINE 33: The management of reproductive health is a widespread health concern. The most important risk factors in the development of disease associated to the reproductive health include: the age of the woman, ovulatory disorders, chromosomal abnormalities, and defective male fertility. A widespread variety of environmental pollutants, such as several classes of pesticides, heavy metals, and air particulate matter plays a key role in the pathogenesis of female infertility. In fact, female fertility and reproductive health are sensitive to toxic exposure, specifically to endocrine disruptor pollutants, and have long-term adverse effects. However, several studies have analyzed the connection between environmental air pollution and female reproductive competence, suggesting an adverse linkage between fertility and toxicants (see, for reference: https://doi.org/10.3390/ijerph17072580). This is the most important flaw of this manuscript: the introductive section of this manuscript needs of a brief and organic premise about reproductive health and its association with the exposure of endocrine disruptors.

LINE 47: BPA has been detected in various body fluids, including urine, serum, seminal plasma, follicular fluid, umbilical cord plasma, and tissues. BPA primarily acts by deregulating the HPG axis, but it also induces oxidative stress, modulates signaling pathways, and causes epigenetic modifications, including DNA methylation, histone modification, and alterations in sperm non-coding RNAs. These changes, including abnormal microRNA (miRNA) expression, may contribute to altered sperm production, fertilization failure, and abnormal embryo development, leading to pregnancy loss (see, for reference: Santiago, J., Simková, M., Silva, J. V., Santos, M. A., Vitku, J., & Fardilha, M. (2024). Bisphenol A Negatively Impacts Human Sperm MicroRNA and Protein Profiles. Exposure and Health, 1-19). Recent studies have also highlighted the association between BPA exposure and the alteration of miRNA expression: in the light of these recent evidence, this manuscript may benefit from provide a brief mention to this topic.

RESULTS:

Figure 1A and 1B: Please, if possible provide in  the figure legend the magnification of the proposed images.

The main topic is interesting, and certainly of great clinical impact. As regards the originality and strengths of this manuscript, this is a significant contribute to the ongoing research on this topic, as it extends the research field on the application of use the 3D gastruloids cell culture platform to study the effects of BPA during gastruloids development, analyzing both the morphogenetic and molecular events involved in their axial elongation. Overall, the contents are rich, and the authors also give their deep insight for some works.

The conclusion of this manuscript is perfectly in line with the main purpose of the paper: the authors have designed and conducted the study properly. As regards the conclusions, they are well written and present an adequate balance between the description of previous findings and the results presented by the authors.

Finally, this manuscript also shows a basic structure, properly divided and looks like very informative on this topic. Furthermore, figures and tables are complete, organized in an organic manner and easy to read.

In conclusion, this manuscript is densely presented and well organized, based on well-synthetized evidence. The authors were lucid in their style of writing, making it easy to read and understand the message, portrayed in the manuscript. However, major concerns of this manuscript are with the introductive section: for these reasons, I have major comments for this section, for improvement before acceptance for publication. The article is accurate and provides relevant information on the topic and I have some major points to make, that may help to improve the quality of the current manuscript and maximize its scientific impact. I would accept this manuscript if the comments are addressed properly.

Author Response

Reviewers’ Comments

Q, your questions; A, our answers (our replies in italics).

Reviewer 2

The main aim and question of this manuscript is to use the 3D gastruloids cell culture platform to study the effects of BPA during gastruloids development, analyzing both the morphogenetic and molecular events involved in their axial elongation.

This manuscript shows rich content, providing a deep insight for some works: the study is within the journal’s scope, and I found it to be well-written, providing sufficient information. Even if the manuscript provides an organic overview, with a densely organized structure and based on well-synthetized evidence, there are some suggestions necessary to make the article complete and fully readable. For these reasons, the manuscript requires major changes.

Please find below an enumerated list of comments on my review of the manuscript:

MINOR POINTS:

Q1: The authors should provide a list of the abbreviations, mentioned in this manuscript.

A1: Although not requested by IJMS, we agree with this reviewer that a list of the abbreviations is useful for the reader. The list has been added at the end of the revised manuscript (page 15).

MAJOR POINTS:

INTRODUCTION:

Q2: LINE 33: The management of reproductive health is a widespread health concern. The most important risk factors in the development of disease associated to the reproductive health include: the age of the woman, ovulatory disorders, chromosomal abnormalities, and defective male fertility. A widespread variety of environmental pollutants, such as several classes of pesticides, heavy metals, and air particulate matter plays a key role in the pathogenesis of female infertility. In fact, female fertility and reproductive health are sensitive to toxic exposure, specifically to endocrine disruptor pollutants, and have long-term adverse effects. However, several studies have analyzed the connection between environmental air pollution and female reproductive competence, suggesting an adverse linkage between fertility and toxicants (see, for reference: https://doi.org/10.3390/ijerph17072580). This is the most important flaw of this manuscript: the introductive section of this manuscript needs of a brief and organic premise about reproductive health and its association with the exposure of endocrine disruptors.

A2: We thank this reviewer for his/her suggestion that contributes to widen the Introduction section.

A new paragraph is present in the revised version of the manuscript (lines 41-46, pages 1-2).

Q3: LINE 47: BPA has been detected in various body fluids, including urine, serum, seminal plasma, follicular fluid, umbilical cord plasma, and tissues. BPA primarily acts by deregulating the HPG axis, but it also induces oxidative stress, modulates signaling pathways, and causes epigenetic modifications, including DNA methylation, histone modification, and alterations in sperm non-coding RNAs. These changes, including abnormal microRNA (miRNA) expression, may contribute to altered sperm production, fertilization failure, and abnormal embryo development, leading to pregnancy loss (see, for reference: Santiago, J., Simková, M., Silva, J. V., Santos, M. A., Vitku, J., & Fardilha, M. (2024). Bisphenol A Negatively Impacts Human Sperm MicroRNA and Protein Profiles. Exposure and Health, 1-19). Recent studies have also highlighted the association between BPA exposure and the alteration of miRNA expression: in the light of these recent evidence, this manuscript may benefit from provide a brief mention to this topic.

A3: Following your suggestion, a brief paragraph has been added in Introduction (lines 41-46, pages 1-2).

RESULTS:

Q4: Figure 1A and 1B: Please, if possible provide in the figure legend the magnification of the proposed images.

A4: We added this information in the legends of Figure 1 and 2.

Q5: The main topic is interesting, and certainly of great clinical impact. As regards the originality and strengths of this manuscript, this is a significant contribute to the ongoing research on this topic, as it extends the research field on the application of use the 3D gastruloids cell culture platform to study the effects of BPA during gastruloids development, analyzing both the morphogenetic and molecular events involved in their axial elongation. Overall, the contents are rich, and the authors also give their deep insight for some works.

The conclusion of this manuscript is perfectly in line with the main purpose of the paper: the authors have designed and conducted the study properly. As regards the conclusions, they are well written and present an adequate balance between the description of previous findings and the results presented by the authors.

Finally, this manuscript also shows a basic structure, properly divided and looks like very informative on this topic. Furthermore, figures and tables are complete, organized in an organic manner and easy to read.

In conclusion, this manuscript is densely presented and well organized, based on well-synthetized evidence. The authors were lucid in their style of writing, making it easy to read and understand the message, portrayed in the manuscript. However, major concerns of this manuscript are with the introductive section: for these reasons, I have major comments for this section, for improvement before acceptance for publication. The article is accurate and provides relevant information on the topic and I have some major points to make, that may help to improve the quality of the current manuscript and maximize its scientific impact. I would accept this manuscript if the comments are addressed properly.

A5: We thank the reviewer for these words of appreciation. We have implemented the Introduction section as requested (see A2 and A3)

Round 2

Reviewer 2 Report

Comments and Suggestions for Authors

The authors have significantly improved the impact and quality of this manuscript.

Author Response

We thank the reviewer for his/her words of appreciation.